Removal of heavy oil from contaminated surfaces with a detergent formulation containing biosurfactants produced by Pseudomonas spp.

http://orcid.org/0000-0002-2380-1770 Farias Charles Bronzo B. 1 2
Soares da Silva Rita de Cássia F. 1 3
Almeida Fabíola Carolina G. 1
Santos Valdemir A. 1 2 3
Sarubbo Leonie A. 1 2 3 leonie.sarubbo@unicap.br
1 Instituto Avançado de Tecnologia e Inovação , RECIFE, PE , Brasil
2 Renorbio, Universidade Federal Rural de Pernambuco , RECIFE, PE , Brasil
3 Escola Icam Tech, Universidade Católica de Pernambuco , RECIFE, PE , Brasil
Chatti Abdelwaheb
Electronic publication date: 2021 Nov 25
Publication date: 2021
Volume: 9
Electronic Location ID: e12518
Received 2021 Aug 14; Accepted 2021 Oct 27
Copyright: © 2021 Farias et al.
Copyright year: 2021
Copyright holder: Farias et al.
License: This is an open access article distributed under the terms of the Creative Commons Attribution License, which permits unrestricted use, distribution, reproduction and adaptation in any medium and for any purpose provided that it is properly attributed. For attribution, the original author(s), title, publication source (PeerJ) and either DOI or URL of the article must be cited.
License URL: https://creativecommons.org/licenses/by/4.0/

Keywords: Biosurfactant, Microbial surfactants, Petroleum, Environmental contamination, Detergents, Pseudomonas

Funding: Research and Development Program of the National Agency of Electrical Energy (ANEEL) Thermoelectric EPASA (Centrais Elétricas da Paraíba) Thermoelectric EPESA (Centrais Elétricas de Pernambuco S.A.) and Termocabo S.A Brazilian fostering agencies Fundação de Amparo à Ciência do Estado de Pernambuco (FACEPE) Coordenação de Aperfeiçoamento de Pessoal de Nível Superior (CAPES) 001 Conselho Nacional de Desenvolvimento Científico e Tecnológico (CNPq) This study was funded by the Research and Development Program of the National Agency of Electrical Energy (ANEEL) and Thermoelectric EPASA (Centrais Elétricas da Paraíba), Thermoelectric EPESA (Centrais Elétricas de Pernambuco S.A.) and Termocabo S.A. This work was also supported by the Brazilian fostering agencies Fundação de Amparo à Ciência do Estado de Pernambuco (FACEPE [State of Pernambuco Science Assistance Foundation]), Coordenação de Aperfeiçoamento de Pessoal de Nível Superior (CAPES [Coordination for the Advancement of Higher Education Personnel]; Finance Code-001) and Conselho Nacional de Desenvolvimento Científico e Tecnológico (CNPq [National Council of Scientific and Technological Development)]. The funders had no role in study design, data collection and analysis, decision to publish, or preparation of the manuscript.

==============================
Industrial plants powered by heavy oil routinely experience problems with leaks in different parts of the system, such as during oil transport, the lubrication of equipment and mechanical failures. The surfactants, degreasing agents and solvents that make up detergents commonly used for cleaning grease-covered surfaces are synthetic, non-biodegradable and toxic, posing risks to the environment as well as the health of workers involved in the cleaning process. To address this problem, surfactant agents of a biodegradable nature and low toxicity, such as microbial surfactants, have been widely studied as an attractive, efficient solution to replace chemical surfactants in decontamination processes. In this work, the bacterial strains Pseudomonas cepacia CCT 6659, Pseudomonas aeruginosa UCP 0992, Pseudomonas aeruginosa ATCC 9027 and Pseudomonas aeruginosa ATCC 10145 were evaluated as biosurfactant producers in media containing different combinations and types of substrates and under different culture conditions. The biosurfactant produced by P. aeruginosa ATCC 10145 cultivated in a mineral medium composed of 5.0% glycerol and 2.0% glucose for 96 h was selected to formulate a biodetergent capable of removing heavy oil. The biosurfactant was able to reduce the surface tension of the medium to 26.40 mN/m, with a yield of approximately 12.00 g/L and a critical micelle concentration of 60.00 mg/L. The biosurfactant emulsified 97.40% and dispersed 98.00% of the motor oil. The detergent formulated with the biosurfactant also exhibited low toxicity in tests involving the microcrustacean Artemia salina and seeds of the vegetable Brassica oleracea. The detergent was compared to commercial formulations and removed 100% of the Special B1 Fuel Oil (OCB1) from different contaminated surfaces, demonstrating potential as a novel green remover with industrial applications.

Introduction

Fuel and lubricant spills that occur in industries during the filling machines and storage tanks and the washing of equipment are one of the causes of the accumulation of petroleum byproducts in the environment. For engines contaminated with lubricating oil, the removal of adhered grease poses a different challenge from contamination generated by oily sludge, as the cleaning process requires the direct application of a detergent, surfactant or solvent, often generating further environmental problems due to the toxicity and accumulation of these substances (Rocha e Silva et al., 2019).

Detergents are generally comprised of a mixture of surfactants (nonionic and anionic, constituting 1%–50%) plus volatile organic solvents, stabilizers, and other additives capable of reducing the surface/interfacial tension of water, though exact compositions are proprietary information (Thomas et al., 2021). For instance, the Corexit dispersants are a mixture of hydrocarbons (60%–100%), organic sulfonic acid salts (10%–30%), and propylene glycol (1%–5%) (Rongsayamanont et al., 2017). Solvents are used to reduce the viscosity of surfactants, to dilute the detergent, to lower its freezing point and to optimize its concentration, while stabilizers help adjust pH and color, stop corrosion, and increase hard water stability of formulations (Dave & Ghaly, 2011).

Detergents are continually released into the environment, as a result of their wide industrial applications, contributing to the pollution of rivers, oceans and soil. Surfactants, which are the main components of detergent formulations, are responsible for foaming in rivers and affect the physicochemical properties of soils, causing harm to the organisms that inhabit these ecosystems due to their prolonged presence in the environment (Drakontis & Amin, 2020; Farias et al., 2021; Silva & Sarubbo, 2021).

Most surfactants in commercial detergents and degreasing formulations are synthesized from petroleum. However, environmental legislation has motivated the development of natural surfactants, such as microbial surfactants, as alternatives to existing products. These biosurfactants are produced from renewable sources and are efficient, non-toxic, biodegradable and stable under extreme environmental conditions (Farias et al., 2021; Markande, Patel & Varjani, 2021).

Many types of biosurfactants have been produced in recent decades, especially those obtained from bacteria of the genus Pseudomonas, which mainly produce glycolipid biosurfactants composed of a hydrophilic head formed by rhamnose molecules and a hydrophobic tail that contains fatty acids (Bezerra et al., 2018). The production of rhamnolipids by Pseudomonas has the advantages of obtaining a high-quality surfactant within a short culture time. However, further processing often poses a problem in the production of rhamnolipids, especially due to the low yields (Płaza & Achal, 2020).

The development of strategies that enable the application of biosurfactants on an industrial scale is of fundamental importance. Such strategies include the selection of adequate substrates, the determination of optimal culture conditions and the improvement of purification processes (Santos et al., 2016). Moreover, depending on the application purpose, the purification step can be eliminated, which substantially reduces the cost of the process. Thus, the oil industry and environmental applications have become greatest markets for these biomolecules, as such applications have minimal purity specifications (Jimoh & Lin, 2019).

Since the cost of production is still a limitation for establishing biosurfactants on the market, they have been blended with various types of cheaper synthetic surfactants. In general, mixed surfactant systems provide superior properties compared to individual surfactants, such as lower interfacial tension (Shah et al., 2019). For example, the aqueous binary system of the Gemini cationic surfactant, ethanediyl-1, 3 bis (dodecyl dimethyl ammonium bromide) (represented as 12-3-12) and the bacterial surfactant Surfactin was able to reduce the interfacial tension of petroelum (Jin et al., 2016). Jian et al. (2011) described that binary mixtures of synthetic surfactants and plant biosurfactants (saponin) showed more foaming capacity and less interfacial activity than the individual components. Song et al. (2013) showed that a mixture of soporolipids and rhamnolipid with polysorbate-80, sorbet-40 and ethylene glycol butyl was able of dispersing crude oil. Bio-based dispersants have also been formulated by mixing biosurfactants with chemical surfactants, as their synergistic behavior can increase pollutant washing efficiency by increasing the pollutant solubility (Zhu et al., 2020; Baharuddin et al., 2020). A bio-based washing agent prepared by mixing 0.3% lipopeptides (an anionic biosurfactant from Bacillus subtilis GY19) and 2% Dehydol LS7TH (a nonionic fatty alcohol ethoxylate oleochemical surfactant) removed 92% of 15% (w/w) petroleum hydrocarbons (Arpornpong et al., 2020).

Although upstream and downstream processes for the production of biosurfactants have been extensively investigated, studies describing the formulation of novel, efficient products containing biosurfactants are still restrict. Therefore, the aim of this work was to investigate the potential application of biosurfactants produced by bacteria of the genus Pseudomonas from previously established substrates and culture conditions in the formulation of a detergent capable of removing heavy oils. The study of a newly formulated washing agent based on the use of microbial surfactants can become the commercial application of these biomolecules more feasible in the near future.

Materials & methods

Microorganisms

The bacterial strains Pseudomonas cepacia CCT6659 (obtained from the culture bank of the André Tosello Research and Technology Foundation located in the city of Campinas, São Paulo, Brazil) Pseudomonas aeruginosa ATCC 9027, Pseudomonas aeruginosa ATCC 10145 and Pseudomonas aeruginosa UCP 0992 (obtained from the culture bank of the Environmental Science Research Center (NPCIAMB) of the Catholic University of Pernambuco, São Paulo, Brazil) were used as biosurfactant producers. The cultures were maintained in test tubes with solid nutritive agar slants under refrigeration at 4 °C.

Preparation of inoculum

For bacterial growth, the CN (nutrient broth) medium was used with the following composition: meat extract (0.1%), yeast extract (0.2%), peptone (0.5%) and sodium chloride (0.5%) at pH 7.0. The growth parameters were temperature of 28 °C and stirring at 150 rpm for 16 h until obtaining an optical density (OD) of 0.7 (corresponding to an inoculum of 107 CFUs/mL) at 600 nm with a concentration of 2.0% (v/v).

Production of biosurfactants

The culture conditions and production media tested for each microorganism were initially established based on previous experiments carried out at our laboratories. The P. cepacia CCT6659, P. aeruginosa UCP 0992, P. aeruginosa ATCC 9027 and P. aeruginosa ATCC 10145 strains were then tested for biosurfactant production using the carbon sources and culture conditions described in Table 1.

Table 1 Media and culture conditions evaluated for production of biosurfactants by strains of Pseudomonas.

Microorganisms	Production media and fermentation conditions	
Carbon sources	Cultivation parameters	
Pseudomonas cepacia CCT6659	2.0% canola frying oil and 3.0% corn steep liquor (Soares da Silva et al., 2017)	pH 7.0; 250 rpm;
60 h; 28 and 37 °C	
5.0% glycerol and 2.0% glucose	pH 7.0; 200 rpm; 96 h; 28 and 37 °C	
1.5% glucose	
2.0% glucose	
3.0% glucose	
Pseudomonas aeruginosa UCP0992	1.5% glucose	pH 7.0; 200 rpm;
96 h; 28 and 37 °C	
2.0% glucose	
3.0% glucose	
2.0% sucrose	
3.0% sucrose	
Pseudomonas aeruginosa
ATCC 9027 and Pseudomonas aeruginosa ATCC 10145	5.0% glycerol and 2.0% glucose	pH 7.0; 200 rpm;
96 h; 28 and 35 °C	
1.0% N-hexadecane and 1.0% glucose	
2.0% sugar cane molasses and 3.0% corn steep liquor	

The biosurfactant from P. cepacia CCT6659 was produced in a mineral medium containing 0.05% KH2PO4, 0.1% K2HPO4, 0.05% MgSO4.7H2O, 0.01% KCl, 0.001% FeSO4.7H2O and 0. 2% NaNO3 supplemented with carbon and nitrogen sources, as described by Soares da Silva et al. (2017). The biosurfactants from P. aeruginosa UCP 0992, P.aeruginosa ATCC 9027 and P. aeruginosa ATCC 10145 were produced in a mineral medium containing 0.1% K2HPO4, 0.1% K2HPO4, 0.02% MgSO4.7H2O, 0.02% CaCl2.H2O and 0.005 % FeCl3.6H2O supplemented with the substrates described in Table 1. Medium surface tensions were between 55–57 mN/m prior to inoculation.

Isolation of biosurfactants

The most promising media and culture conditions based on the best surface tension values were selected for isolation of the biosurfactants, which were extracted with the addition of ethyl acetate (C4H8O2) to the broth obtained after fermentation (1:1, v/v). The mixture was centrifuged at 5,000 rpm for 10 min, stirred vigorously for 15 min and allowed to stand until phase separation. The organic phase was removed, and the operation was repeated twice more. The product was concentrated from the combined organic phases, which were evaporated at 40 °C in a rotary evaporator. A viscous yellowish product was obtained, which was treated with NaOH and washed with acetone for maximum removal of impurities and dirt. The yield of the isolated biosurfactant was expressed as g/L.

Determination of surface tension and critical micelle concentration

Surface tension was measured in an automatic tensiometer (KSV Sigma 700, Espoo, Finland) using a Du Noüy ring. For such, the platinum ring was immersed in dilutions of the isolated biosurfactants (0.1 g/L) in distilled water and the force required to pull the ring through the air-liquid interface was recorded. The critical micelle concentration (CMC) of the previously isolated biosurfactants was determined by measuring the surface tension of a water sample during the gradual addition of the surfactant until reaching a constant tension value. The CMC was expressed as mg/L of the isolated surfactants.

Determination of emulsification index

To determine the emulsification activity of the biosurfactants, samples of cell-free broth (crude biosurfactant) were analyzed based on Cooper & Goldenberg (1987). For such, 3.0 mL of a hydrophobic compound (soybean oil, corn oil or motor oil) and 3.0 mL of the biosurfactant solution were placed in a test tube and the mixture was vortexed for 1 min. After 24 h, the percentage of the emulsion was calculated as follows:

(1) E=100×EP/H

in which EP is the height of the emulsified phase and H is the total height of the mixture (both expressed in centimeters). Emulsifying activity prior to inoculation was used as the negative control for each culture medium. All analyses were performed with four experiments.

Dispersion of hydrophobic contaminant

The ability of the biosurfactants to disperse an oil slick was simulated in the laboratory by contaminating water samples with 5% motor oil. The crude biosurfactants (cell-free broth) were added at proportions of 1:2, 1:8 and 1:25 (vol/vol) of biosurfactant/oil. Negative controls were carried out with distilled water. The results were observed visually and the relation between the required volume of biosurfactant and dimensions of the formed halo was calculated (Saeki et al., 2009).

Maximization of selected biosurfactant extraction

Based on the results of the previous experiments involving the determination of surface tension, the emulsification index and oil dispersion, only one biosurfactant (i.e., that produced by P. aeruginosa ATCC 10145 cultured in mineral medium composed of 5.0% glycerol and 2.0% glucose for 96 h) was selected for the subsequent experiments. Three biosurfactant extraction and isolation processes (described below) were evaluated considering the increase in yield obtained with few downstream steps.

Chloroform/methanol system: For isolation of the biosurfactant, the pH of the cell-free broth was initially adjusted to 2.0 with a 6 M solution of HCl. The same volume of chloroform/methanol (2:1, v/v) was added. The mixture was stirred vigorously for 15 min and left to stand for phase separation. The organic phase was removed, and the operation was repeated twice more. The product was concentrated from the collected organic phases using a rotary evaporator (Silva et al., 2010).

Extraction by acid precipitation: A 2N HCl solution was used to acidify the supernatant containing the biosurfactant until reaching pH 2. The mixture was then incubated for 24 h at 4 °C. The formed precipitate was collected by centrifugation at 10,000 rpm and 4 °C for 30 min. The precipitate was then subjected to a drying process for 24 h and then separated for subsequent tests (Shah et al., 2016).

Foam fractionation isolation: A simple fractionation system (Fig. 1) was used to recover the biosurfactant contained in the foam. In this system, the cell-free broth was fed into the top of the flask with the aid of a peristaltic pump (Winterburn, Russell & Martin, 2011). The foam was collected in another reservoir, forming a concentrated biosurfactant solution after coalescence. Foam fractionation was performed for 1 h to ensure stable foam production conditions and the sample was collected at the end of the process. Foam fractionation performance was measured in terms of enrichment and biosurfactant recovery standards, which are defined in the following equations:

Figure 1 Illustration of foam fractionation method. Air inlet and foam formation from P. aeruginosa ATCC 10145 cell-free broth (1); foam transport to receiving flask (2); foam coalescence and air outlet (3).

Enrichment=CfCi

Recovery=CfVfCiVi×100

in which Cf is the concentration of biosurfactant in the foamate, Ci is the coincident biosurfactant concentration in the feeder, Vi is the initial liquid volume and Vf is the volume of foamate collected. To compare the yield obtained with this method, biosurfactant concentration in the foamate was determined through chloroform/methanol extraction.

Application of biosurfactant in formulation of industrial detergent

After isolation, the selected biosurfactant was used as the main component of a detergent. To define the most promising formulation, different formulations were initially investigated using natural components. Biodegradable, non-toxic compounds were selected based on previous studies (Rocha e Silva et al., 2020) for the formulation of a detergent capable of removing heavy oil. The formulation consisted of a natural organic solvent to maximize the fluidization of the heavy oil, a thickening fatty alcohol to increase the viscosity of the formula, a gum as an emulsion stabilizer and the microbial biosurfactant for the removal the oily fraction during the surface cleaning process. Previously, the formulation components, i.e., the isolated biosurfactant (10%), the natural organic solvent and the thickening fatty alcohol (10%), were tested (30 mL) separately for oil removal to ensure the viability of the formulation containing the biosurfactant, as it is described below using the surface of a glass slide of known mass contaminated with 100 µL of heavy oil.

Tests were then conducted to determine the best percentage of each component of the formula. The mixing of all components was performed in a mechanical stirrer (Tecnal LTDA, São Paulo, Brazil) at 2,000 rpm for 30 min at 80 °C. The formation of phases was followed visually over the course of a month. The purpose of quantifying the compounds was to obtain a stable emulsion with the smallest possible amount of each component. All proportions of the compounds are detailed in Table 2. The formulations were prepared by dissolving the solid components and adding the solvent to complete 100% of the total mixture.

Table 2 Compounds evaluated for formulation of industrial detergent containing biosurfactant.

Components	Quantities (%)	
Natural organic solvent (v/v)	20.0 and 30.0	
Isolated biosurfactant (w/v)	0.5 and 1.0	
Thickener fatty alcohol (w/v)	0.5, 1.5, 2.0, 2.5 and 3.0	
Stabilizing gum (w/v)	0.4, 0.5 and 0.6	

The efficiency of the formulations was determined based on the stability of the emulsions (no phase separation) and the removal of heavy oil impregnated on different surfaces, as described below. The formulation with the smallest amounts of the components, best stability and greatest oil removal was selected for tests on different surfaces. The selected formulation was subjected to comparative tests with four synthetic commercial detergents of industrial use (identified as commercial detergent A, B, C and D) for cleaning parts, floors and machinery impregnated with heavy oil. Detergent A is formulated with sodium silicates, amines and alcohols (pH 12.0 to 14.0). Detergent B has phosphoric acid and paint strippers (pH 1.0 to 4.0). Detergent C is formulated with ammonium hydroxide and detergent D is a set of hydrogenated (aliphatic and naphthenic) alkaline solvents with high dielectric strength.

Assessment of toxicity of industrial detergent against Artemia salina

The toxicity test was conducted using microcrustacean (Artemia salina) larvae as the toxicity indicator with test solutions of the industrial detergent diluted at proportions of 1:5 and 1:10 (v/v) prepared in distilled water and used at concentrations of 1% and 2%. Larvae were used within one day of hatching. The tests were conducted in 15-ml penicillin tubes containing 10 shrimp larvae in 10 ml of seawater. The larvae were observed for 24 h and mortality was calculated (Meyer et al., 1982). Seawater without biosurfactant was used as the control. Each test was run in triplicate.

Assessment of phytotoxicity of industrial detergent

Phytotoxicity of the detergent was evaluated in a static assay involving the cabbage species Brassica oleracea (var. capitata), with the analysis of seed germination and root growth, based on Tiquia, Tam & Hodgkiss (1996). Test solutions of diluted detergent at proportions of 1:5 and 1:10 (v/v) were prepared in distilled water at concentrations of 1% and 2%. Toxicity was determined in sterile Petri dishes (10 cm) containing Whatman No. 1 filter paper disks. Ten seeds (previously treated with NaClO) were placed symmetrically in the dish, inoculated with 5 mL of the test solution and maintained at 28 °C for 5 days. Distilled water was used as a control. After incubation in the dark, seed germination, root growth (≥5 mm) and the germination index (GI) were calculated according to the formulas below:

Relative seed germination (%)= (number of seeds germinated in extract/number of seeds germinated in control)×100

Relative root length (%)= (mean root length in extract/mean root length in control)×100

GI=[(%seedgermination)×(%rootgrowth)]/100%.

The mean and standard deviation of triplicate samples of each concentration were calculated.

Application of detergent in cleaning of surfaces contaminated with heavy oil

The detergent formulated with the isolated biosurfactant was evaluated for heavy oil removal from different types of contaminated surfaces, i.e., smooth surface (glass slide), metallic surface (threaded nuts) and plastic surface (oil storage container). The heavy oil was obtained from a thermoelectric power plant and classified as B1 Special OCB1 fuel oil (PETROBRAS, Rio de Janeiro, Brazil). This oil is a complex mixture of hydrocarbons. Its kinematic viscosity at 60 °C is 620 Cst, its flash point 66 °C and its density at 20 °C 0.968 g/mL. Part of the surface of a glass slide of known mass was uniformly contaminated with 100 µL of heavy oil. Metal nuts were completely covered with OCB1 oil. Part of the outer surface of a plastic container was contaminated with oil, simulating an overflow of stored petroleum byproducts. The tests were performed statically (pieces immersed in detergent at rest for 3 min). For the storage container, 100 mL of the detergent was manually spread on the surface and the oil was removed with the aid of absorbent material (sponge, paper towel, etc.). In this case, the percentage of removal was determined gravimetrically and visually. For the glass and metal, the specimens were oven dried at 40 °C for 30 min and the respective weights were recorded. The removal rate was calculated as follows:

I=100×((Wc−Ww))/((Wc−Wi))

in which Wc is the weight of the contaminated test specimen, Ww is weight of the test specimen after washing and Wi is the initial weight of the test specimen.

Statistical analysis

The data were submitted to statistical analysis using the one-way procedure in Statistica® (version 7.0), followed by linear one-way analysis of variance (ANOVA). All triplicate results were expressed as mean ± standard deviation. Differences were examined using Tukey’s post hoc test, with a 95% significance level.

Results

Properties of biosurfactants

In this work, bacteria capable of producing surfactants with the ability to mobilize and remove medium and heavy oils used at industrial plants were studied. The properties of the biosurfactants were evaluated and the reduction in the surface tension of the medium was the main parameter for the selection of surfactants.

Reduction in surface tension

The results obtained for biosurfactants produced by P. cepacia CCT 6659 in different media and culture conditions are shown in Fig. 2.

Figure 2 Surface tensions after culture of P. cepacia CCT6659 (60 h, 250 rpm) at 28 and 37 °C (Soares da Silva et al., 2013) and at 28 and 37 °C (96 h, 200 rpm) in glycerol and glucose (A) and in glucose (B).

Data presented are the average of triplicate experiments and error bars indicate standard deviation around the mean (***0.2–0.5; **0.6–1.4).

The culture medium described by Soares da Silva et al. (2017) was used as a comparison to the media evaluated in the present study for the same strain, since P. cepacia proved to be an excellent biosurfactant producer in the study cited, achieving surface tension around 26 mN/m.

The purpose of this initial stage was to study the most suitable culture media for the production of microbial surfactants considering the technical-economic feasibility of large-scale production and with the aim of increasing the yield with few downstream steps. The selection of the most favorable fermentation parameters for the production process was based on the determination of surface tension. For the biosurfactant obtained from the bacterium P. cepacia in the culture media analyzed, temperature exerted no significant influence on fermentation, although more satisfactory surface tension results were obtained at 37 °C. However, the most promising result was obtained with the highest concentration of glucose (3.0%) at 28 °C, with which surface tension of 31.60 mN/m was reached.

Figure 3 shows the results of the analysis of the biosurfactants produced by P. aeruginosa UCP 0992.

Figure 3 Surface tensions obtained after culture of P. aeruginosa UCP 0992. Fermentations performed for 96 h at 200 rpm at 28 and 37 °C in media containing sucrose (A) and glucose (B).

Data presented are the average of triplicate experiments and error bars indicate standard deviation around the mean (***0.2–0.5).

Temperature also did not exert an influence on the fermentation of P. aeruginosa UCP 0992. Lower concentrations of glucose and sucrose in the culture enabled a greater reduction in surface tension. The use of lower concentrations of substrates translates to an increase in the economy of the process due to the reduction in input costs, which is essential to increasing the market competitiveness of surfactants.

Figure 4 shows the surface tension reduction capacity of biosurfactants produced by the strains P. aeruginosa ATCC 10145 and P. aeruginosa ATCC 9027 at temperatures of 28 and 35 °C. The mineral medium containing glycerol and glucose as carbon sources at the two temperatures evaluated was the most promising for the production of biosurfactants. As no significant difference was found between temperatures of 28 and 35 °C during the 96 h of culture, 28 °C was considered more viable for obtaining biosurfactants due to the reduction in electricity costs, making the process more feasible.

Figure 4 Surface tensions after culture of P. aeruginosa ATCC 10145 and P. aeruginosa ATCC 9027 in glycerol plus glucose, hexadecane plus glucose and molasses plus corn steep liquor at 28 °C (A) and 35 °C (B).

Data presented are the average of triplicate experiments and error bars indicate standard deviation around the mean (***0.2–0.5; **0.6–1.4).

The surface tension results obtained for the four microorganisms analyzed showed that the strains P. aeruginosa ATCC 10145 and P. aeruginosa ATCC 9027 achieved the best performance when cultivated in the medium containing 5% glycerol and 2% glucose carbon sources for 96 h at 28 °C, reaching excellent surface tension values of around 25.00 and 29.00, respectively. Thus, the biosurfactants produced by these two strains were selected for the evaluation of emulsification and dispersion capacity regarding hydrophobic compounds.

Emulsification index

The results of the emulsification activity tests of biosurfactants from P. aeruginosa ATCC 10145 and P. aeruginosa ATCC 9027 cultivated in the medium containing 5% glycerol and 2% glucose for different hydrophobic compounds can be seen in Fig. 5.

Figure 5 Emulsification capacity of biosurfactants from P. aeruginosa ATCC 10145 and P. aeruginosa ATCC 9027 in medium composed of 5.0% glycerol and 2.0% glucose for 96 h at 28 °C with different oils.

Data presented are the average of triplicate experiments and error bars indicate standard deviation around the mean (***0.2–0.5; **0.6–1.4; *1.5–2.0).

The biosurfactants from both strains exhibited good emulsification capacity for soybean, corn and motor oils, although with specific affinities for the different hydrophobic compounds evaluated. In general, the biosurfactants exhibited good emulsification efficiency for denser oils, especially motor oil, which is often used in industrial engines. The biosurfactant produced by P. aeruginosa ATCC 10145 emulsified 97.50% of the motor oil. Furthermore, the emulsions had relatively constant indices in relation to other less dense hydrophobic compounds, forming emulsions that remained stable in the long term.

Dispersion capacity

The dispersion capacity of microbial surfactants obtained from the strains P. aeruginosa ATCC 10145 and P. aeruginosa ATCC 9027 cultivated in the medium containing glycerol and glucose is shown in Fig. 6.

Figure 6 Dispersion of motor oil by biosurfactants from P. aeruginosa ATCC 10145 and P. aeruginosa ATCC 9027 cultivated in 5.0% glycerol and 2.0% glucose for 96 h at 28 °C.

Data presented are the average of triplicate experiments and error bars indicate standard deviation around the mean (***0.2–0.5; **0.6–1.4).

The data demonstrated excellent dispersant activity. All biosurfactant/motor oil ratios tested were promising, especially the biosurfactant from P. aeruginosa ATCC 10145, which exhibited dispersion rates of approximately 99.00%. The results indicate excellent interaction between the biosurfactants and oil, even at oil concentrations 25-fold higher than that of the surfactants.

Critical micelle concentration

Another important parameter for evaluating the efficiency of a biosurfactant is the CMC, which is the minimum concentration of biosurfactant necessary for the maximum reduction in surface tension. In the present study, the biosurfactants isolated from the P. aeruginosa ATCC 10145 and P. aeruginosa ATCC 9027 strains exhibited excellent surfactant properties, yields and CMCs (Table 3 and Fig. 7).

Figure 7 Critical micelle concentration plots of the biosurfactants produced from (A) P. aeruginosa ATCC 10145 and (B) P. aeruginosa ATCC 9027 cultivated in 5.0% glycerol and 2.0% glucose for 96 h at 28 °C.

Table 3 Properties of biosurfactants from P. aeruginosa ATCC 10145 and P. aeruginosa ATCC 9027 cultivated in medium composed of 5.0% glycerol and 2.0% glucose for 96 h at 28 °C.

Microorganisms	Surface tension (mN/m)	CMC (mg/L)	Yield (g/L)	
Pseudomonas aeruginosa ATCC 9027	28.00 ± 0.37	20.00	1.20 ± 0.08	
Pseudomonas aeruginosa ATCC 10145	26.40 ± 0.32	60.00	11.97 ± 0.29	

The yields are related to surfactants with low purity specifications, as industries that employ heavy oils does not require surfactants with a high degree of purification. Thus, downstream steps, which represent nearly 60% of total production costs, can be eliminated, making the use of biosurfactants economically advantageous (Santos et al., 2016).

The data demonstrate the considerable potential of the biosurfactants evaluated for oil removal purposes. From the results considering surface tension, the emulsification index and oil dispersion, P. aeruginosa ATCC 10145 was selected as the best biosurfactant producer. Thus, the biosurfactant produced by this bacterial strain was used as one of the main components in the formulation of a biodegradable detergent for application at industrial plants. In the culture medium containing 5.0% glycerol and 2.0% glucose as carbon sources, the biosurfactant from P. aeruginosa ATCC 10145 promoted intense foam formation (Figs. 8A and 8B), indicating considerable emulsifying capacity.

Figure 8 Foam formation promoted by biosurfactant from P. aeruginosa ATCC 10145 cultivated in 5.0% glycerol and 2.0% glucose for 96 h at 28 °C. Crude biosurfactant (A and B) and isolated biosurfactant (C).

Based on the results obtained, the biosurfactant produced from the selected strain under its best culture conditions was isolated, as described above. The yield was approximately 12 g/L. An increase in this yield can be achieved by optimizing the production parameters and studying upstream and downstream technologies for this biomolecule. The isolated biosurfactant was able to reduce the surface tension of water from 71 mN/m to 28 mN/m and form a very stable foam even when diluted (Fig. 8).

Isolation methods of selected biosurfactant

The surface tension and yield of the biosurfactant from P. aeruginosa ATCC 10145 with different extraction methods results are described in Table 4. The best reduction in surface tension of the cell-free broth and the liquid was achieved with the foam fractionation process. The best yield was achieved with foam fractionation and extraction using the chloroform/methanol system. Considering the need to obtain a low-cost surfactant with few downstream steps, the use of the broth and coalesced liquid proved to be more promising.

Table 4 Surface tension and yield of biosurfactant from P. aeruginosa ATCC 10145 cultivated in medium composed of 5.0% glycerol and 2.0% glucose with different extraction methods.

Biosurfactant and extraction methods	Surface tension (mN/m)	Yield (g/L)	
Cell-free broth (crude biosurfactant)	24.06 ± 0.22	-	
Biosurfactant isolated with chloroform/methanol	30.60 ± 0.13	1.82 ± 0.02	
Biosurfactant isolated by acid precipitation	29.43 ± 0.15	0.44 ± 0.00	
Biosurfactant isolated by foam fractionation	26.50 ± 0.22	11.45 ± 0.21%*	
Note:

* % v/v

Assessment of toxicity of industrial detergent

Ecotoxicological tests using cabbage (Brassica oleracea) seeds and microcrustacean (Artemia salina) larvae as indicators demonstrated the absence of toxicity of the detergent formulated with the biosurfactant (Table 5). When compared to commercial detergents used in thermoelectric plants, the formulated detergent composed of 20% natural solvent, 2.0% thickening fatty alcohol, 0.5% stabilizing gum and 0.5% biosurfactant presented the same oil removal efficiency with no toxicity, which is a considerable advantage over commercial detergents. It is evident, therefore, that the biodetergent offers better working conditions for cleaning workers, ensuring a healthier environment due to the use of a harmless product.

Table 5 Toxicity tests of detergent formulated with 20% natural solvent, 2.0% thickening fatty alcohol, 0.5% stabilizing gum and 0.5% biosurfactant from P. aeruginosa ATCC 10145.

Detergent/water ratio (v/v)	Mortality of Artemia salina larvae (%)	Brassica oleracea seed germination (%)	
1/5	10.00 ± 0.01	100.00 ± 0.01	
1/10	No mortality	100.00 ± 0.00	

Application of industrial detergent in cleaning of surfaces contaminated with heavy oil

Since the detergent formulation has several components, they were previously tested separately for oil removal. These controls were carried out to show their effect on the oil removal efficiency. Table 6 shows percentages of oil the removed by the detergent components. It can be observed that the isolated biosurfactant has a similar efficiency when compared to the organic solvent.

Table 6 Removal of heavy oil (OCB1) from glass slide by the components of detergent formulated with biosurfactant from P. aeruginosa ATCC 10145 cultivated in 5.0% glycerol and 2.0% glucose for 96 h at 28 °C.

Formulation components	Oil removal	
Natural organic solvent	40 ± 0.51%	
Isolated biosurfactant	40 ± 0.37%	
Thickener fatty alcohol	25 ± 0.13%	

After testing each component separately, different components concentrations were evaluated. Among the various component concentrations studied, the best formulation was that containing 20% natural solvent, 2.0% thickening fatty alcohol, 0.5% stabilizing gum and 0.5% biosurfactant. This combination of compounds formed a stable emulsion over the evaluation time (1 month) capable of removing heavy oil (OCB1) from different surfaces in a short time.

The detergent formulation was able to remove 100% of the heavy oil from a glass slide, threaded nut and plastic surface (Figs. 9 and 10). The results indicate the benefit of the detergent and that this formulation has potential for future use in the cleaning of parts, machines and equipment impregnated with oils, greases and petroleum byproducts, such as OCB1 oil, commonly used in industrial plants. The green detergent could be a good solution for industrial cleaning, replacing the conventional products and solvents commonly used in such environments.

Figure 9 Removal of heavy oil (OCB1) from contaminated surface by industrial detergent formulated with the biosurfactant from P. aeruginosa ATCC 10145 cultivated in 5.0% glycerol and 2.0% glucose.

(A) Heavy oil impregnated glass slide for detergent immersion. (B) Oil-impregnated slide immersed in detergent. (C) Immersion in distilled water for complete removal of destabilized oil from glass surface. (D) Complete oil removal.

Figure 10 Removal of heavy oil (OCB1) from metallic and plastic surfaces by industrial detergent formulated with biosurfactant from P. aeruginosa ATCC 10145 cultivated in 5.0% glycerol and 2.0% glucose.

I-Metallic surface. (A) Threaded nuts impregnated with heavy oil OCB1 for immersion in detergent threaded. (B) Nuts impregnated with oil immersed in detergent. (C) Immersion in distilled water for complete removal of destabilized oil from metal surface. (D) Complete oil removal. II-Plastic surface. (A) OCB1 oil-impregnated plastic storage container for detergent application. (B) Application of detergent to oil-impregnated surface. (C) Complete oil removal.

The selected formulation was also subjected to comparative tests with commercial detergents used in industrial plants, achieving very satisfactory results in the removal of OCB1 oil from metal surfaces through processes of solubilization and mobilization. Table 7 presents the results of the comparison of commercial detergents and the detergent formulated with the biosurfactant.

Table 7 Removal of heavy oil from metal surfaces by commercial detergents and detergent formulated with biosurfactant from P. aeruginosa ATCC10145 cultivated in 5.0% glycerol and 2.0% glucose for 96 h at 28 °C.

Detergents	Removal of OCB1 heavy oil (%)	
Water (control)	2.10 ± 0.05	
Commercial detergent A*	17.30 ± 0.06	
Commercial detergent B*	10.20 ± 0.10	
Commercial detergent C*	42.50 ± 0.15	
Commercial detergent D*	100.00 ± 0.05	
Detergent formulated with biosurfactant	100.00 ± 0.01	
Note:

* Commercial detergent A: formulated with sodium silicates, amines and alcohols; Commercial detergent B: formulated with phosphoric acid and paint strippers; Commercial detergent C: formulated with ammonium hydroxide; Commercial detergent D formulated with set of hydrogenated (aliphatic and naphthenic) alkaline solvents with high dielectric strength.

The data demonstrated the potential of the green detergent for use in the removal of high-density oil, such as OCB1 oil (heavy oil used as fuel for power generation at thermoelectric plants), the high viscosity of which constitutes an obstacle to the action of commercial detergents/mixtures. Although commercial detergent D achieved the same removal rate (100%), one must bear in mind the considerable corrosive and volatile capacity of the product, which can harm the health of operators and limit the durability of metal parts. In contrast, the detergent formulated with the biosurfactant also achieved 100% removal of the OCB1 oil.

Discussion

Biosurfactant-producing bacteria are found in a wide variety of habitats from aquatic environments (fresh water, seawater and groundwater) to terrestrial environments (soil, sediment and silt). Environmental conditions exert a direct influence on the type of biosurfactants that microorganisms produce (Decho & Gutierrez, 2017; Nikolova & Gutierrez, 2021).

Bacteria of the genus Pseudomonas are able to produce different kinds of biosurfactants, including glycolipids (rhamnolipids) and lipopeptides. The most widely studied biosurfactant producer is P. aeruginosa, which produces rhamnolipids that form stable emulsions with hydrocarbons (Bollinger et al., 2020; Shahaliyan, Safahieh & Abyar, 2015).

The production of biosurfactants by microbial species using renewable substrates and optimizing culture conditions (fermentation time, agitation speed, pH of the medium and nutrients) enables obtaining compounds with distinct properties that make these natural surfactants comparable or even superior to their synthetic counterparts (Silva et al., 2014). Indeed, the potential of bacteria of the genus Pseudomonas combined with the use of different media and fermentation conditions has enabled obtaining extremely promising biosurfactants in terms of surface tension reduction, dispersion and emulsifying capacity, which are fundamental properties in the selection of the biomolecule to compose a formulation with industrial purposes.

The reduction in surface or interfacial tension is considered the main parameter for detecting a surface-active compound in a given production medium. This attraction force is the result of the joint interaction between the molecules of liquids. A potent biosurfactant decreases surface tension as its concentration increases in the aqueous medium, enabling greater interaction between immiscible liquids, such as oil and water (Silva et al., 2014). According to Eslami, Hajfarajollah & Bazsefidparc (2020), a more than 8 mN/m reduction in surface tension is needed to identify a microorganism as a biosurfactant producer. However, rhamnolipids can reduce the surface tension of water from 72 to less than 30 mN/m. In the present study, it was possible to obtain a biosurfactant with excellent surface tension reduction capacity through the different media and culture conditions tested with four bacterial strains, achieving values around 26.40 mN/m.

The literature offers several studies on biosurfactants produced by species of Pseudomonas with similar properties to those obtained in this research. For instance, Patowary et al. (2017) produced a biosurfactant from the strain P. aeruginosa PG1 isolated from soil contaminated with hydrocarbons that reduced the surface tension of the culture medium from 51.80 to 29.60 mN/m. Câmara et al. (2019) also obtained a biosurfactant from P. aeruginosa able to reduce the surface tension of water from 72.00 to 35.26 mN/m. Using the strain P. aeruginosa NCIM 5514 cultivated in a medium with crude oil as the carbon source, Varjani & Upasani (2019) produced a biosurfactant able to reduce the surface tension to 29.14 ± 0.05 mN/m.

Most microbial surfactants are substrate specific, solubilizing or emulsifying different hydrocarbons at different rates. The inability to emulsify some hydrocarbons may be due to the inability of the biosurfactant to stabilize microscopic droplets (Bouassida et al., 2018). In this work, the biosurfactant from P. aeruginosa ATCC 10145 exhibited considerable affinity with motor oil, as demonstrated by the emulsification rates higher than 97% obtained for this substrate. In contrast, the vegetable oils tested were not emulsified to the same degree, although 50% of these oils were also stably emulsified by the biomolecule. This affinity with motor oil is related to the chemical structure of the biosurfactants evaluated (Sousa et al., 2014), which enables greater intermolecular interaction with the chemical structure of the oil, making these microbial surfactants suitable for applications in industrial environments that generate mineral oil waste (Almeida et al., 2016). Anaukwu, Ogbukagu & Ekwealor (2020) described the optimization of biosurfactant production by P. aeruginosa to determine the impact of different renewable waste products. The preferred carbon source of the bacterial isolate was sugarcane molasses. The biosurfactant achieved an emulsification index of 96.3% ± 0.75%. In the study by Bezerra et al. (2020), the emulsification index of a biosurfactant from P. aeruginosa reached 71.0% for vegetable oils.

Another important parameter for evaluating the efficiency of a biosurfactant is the CMC, which is the minimum concentration of biosurfactant necessary for the maximum reduction in surface tension. The increase in the concentration of a surfactant in the medium promotes the formation of micelles, which are aggregated amphipathic molecules with the hydrophilic portion positioned outwards and hydrophobic portion positioned inwards, enabling the trapping of oil in the micellar complex and promoting emulsification (Arsene et al., 2021). The CMC is another important parameter to consider in the development of large-scale production processes, as it enables the prediction of efficiency and economic viability (Santos et al., 2016). The CMC found for the biosurfactants from P. aeruginosa ATCC 10145 and ATCC 9027 were higher than many values described in the literature for potent bacterial surfactants, such as the biosurfactant produced by P. aeruginosa S5, which had a CMC of 96.50 mg/L and reduced surface tension from 72.20 to 29.60 mN/m (Sun et al., 2019). Similar results were also found in the work by Anaukwu, Ogbukagu & Ekwealor (2020), with the surface tension of distilled water reduced from 72.10 mN/m to 35.00 ± 0.0 mN/m and a CMC of 60.00 mg/L.

Although biosurfactant isolation techniques still need to be improved, the yields obtained for P. aeruginosa biosurfactants in this work were satisfactory when compared to those described in the literature for other microbial biosurfactants. The yields obtained are related to surfactants with low purity specifications, as industries that employ heavy oils does not require surfactants with a high degree of purification. Thus, downstream steps, which represent nearly 60% of total production costs, can be eliminated, making the use of biosurfactants economically advantageous.

An oil dispersant consists of a mixture of various chemicals, but the major constituents are surfactants, which are responsible for solubilization and dispersion. Due to their amphipathic structure, surfactants solubilize oil through the formation of micelles, which disperse in water (Silva et al., 2014). Corexit is one of the most widely used chemical dispersants for oil solubilization. This chemical agent passed initial testing by the US Environmental Protection Agency (EPA) after the government asked to cut its use in half. For the test, the EPA analyzed eight chemicals to determine how oil dispersants affect marine wildlife. All products had negative effects despite the favorable dispersant performance. The EPA still permits the use of Corexit but stresses the need to find a more environmentally friendly alternative (Silva et al., 2014). In this scenario, biosurfactants constitute a promising alternative to replace chemical dispersants in formulations. The dispersion results obtained in the present study for biosurfactants from the strains P. aeruginosa ATCC 10145 and ATCC 9027 were above 70% regardless of the type of oil and dilution tested and reached rates of up to 95%, demonstrating the capacity of these biomolecules as dispersing agents. Other biosurfactants produced by species of Pseudomonas have proven to be highly efficient dispersants of petroleum byproducts. Soares da Silva et al. (2017) demonstrated that the biosurfactant from P. cepacia dispersed oil (81%), indicating the potential of the biosurfactant for application in the control of oil spills. Sun et al. (2019) proved that the glycolipid biosurfactant produced by a strain of P. aeruginosa was able to disperse approximately 74% of petroleum byproduct in water. A new green detergent similar to the formulation described in this work, composed by 1.0% of the biosurfactant from Starmerella bombicola ATCC 22214, 0.4% of hydroxyethyl cellulose, 1.0% of EDTA and 0.2% of potassium sorbate as preservative was tested in the remediation of soils contaminated with hydrocarbons (Silva et al., 2021). The formulation showed effectiveness in removing motor oil from contaminated sandy soil (80.0%) and beach sand (65.0%) under static conditions. Although the conditions applied were totally different form the experiments carried out in our research, the static condition can be compared to the immersion tests carried out for heavy oil removal and shows, once more, the feasibility of detergent formulations based on biosurfactants.

Studies have reported that some forms of extraction and isolation are more adequate for specific biosurfactants. In the present study, three forms of isolation were evaluated for the biosurfactant from P. aeruginosa ATCC 10145 to select the most economically viable extraction method capable of obtaining the highest concentration of biosurfactant for application in the formulation of the detergent. The most promising isolation method was foam fractionation. Several studies in the literature report the characteristics of the foaming method of biosurfactants in fermentation broths and the use of foaming for product recovery. Beuker et al. (2016) evaluated the efficacy of foam fractionation in rhamnolipid production. The foam fractionation process was easily manageable in an enriched bioreactor and the biosurfactants were highly concentrated in the foam during the culture process, with an increase in rhamnolipid recovery of up to 97%. Bages-Estopaa et al. (2018) used the foaming method for trehalolipid biosurfactants in fermentation broths. Trehalolipids were produced by the bacterium Rhodococcus sp. PML026 with hexadecane as carbon source. Hexadecane suppresses foaming during fermentation, which is beneficial during the growth and production phases. In the study, the hexadecane substrate was exhausted at the end of the fermentation, enabling the formation of foam and the separation of the product by foam fractionation, with a total trehalolipid recovery rate of 23% to 58%. Blesken et al. (2020) proposed this method as a pre-purification step. Foam fractionation was coupled to the bioreactor operation using an external fractionation column to decouple the production of a rhamnolipid by P. putida KT2440. The technique enabled continuous separation of the surfactant, which is particularly suitable for scale-up. Surfactant concentrations of 7.50 g/L were obtained in the fractionated foam. These studies demonstrate a strategic, low cost, eco-friendly technology for separating biosurfactants from emulsified fermentation broths.

The absence of toxicity is of fundamental importance for the application of a product in industrial sectors. Detergents and surfactants are among the most widely used chemicals in industries. The toxic effect of a detergent depends on its mode of action, the toxicity of the active ingredient (surfactant) and the response of organisms. Short-term ecotoxicity bioassays are analytical methods that enable the assessment of bioavailability and the toxicity of chemical substances by showing acute effects on the survival or mobility of test organisms. The toxicity tests conducted in the present study revealed the non-toxic nature of the detergent formulated with the P. aeruginosa biosurfactant ATCC1045, demonstrating the viability of this novel product in the market. The detergent formulation produced in this work also achieved better results in comparison to microbial biosurfactants described in the literature (Ostendorf et al., 2019; Rocha e Silva et al., 2014).

The literature describes several studies conducted to determine the toxicity of chemical detergents on test organisms, but little is known about the toxic effects of biobased detergents, which are considered to be environmentally friendly. Uc-Peraza & Delgado-Blas (2015) studied the toxicity of three commercial detergent formulations (ROMA®, FOCA® and BLANCA NIEVES®) to determine the median lethal concentration (LC50). FOCA® was the most toxic detergent, followed by BLANCA NIEVES® and ROMA®. According to the authors,the difference in toxicity among the three detergents can be related to the concentrations of the anionic surfactants or to other ingredients such as sodium silicate, enzymes, sodium tripolyphosphate, bleaches and perfumes. In acute toxicity tests with fish, Baharuddin et al. (2020) found that a dispersant based on water and ionic liquids (ILs) [1-butyl-3-methylimidazolium lauroylsarcosinate, 1,1′-(1,4-butanediyl)bis (1-H-pyrrolidinium) dodecylbenzenesulfonate, tetrabutylammonium citrate, tetrabutylammonium polyphosphate and tetrabutylammonium ethoxylate oleyl ether glycolate] was practically non-toxic and exhibited excellent biodegradability throughout the test period. This novel surfactant was considered an oil facilitator and environmentally safe for use in oil spill remediation process and could effectively replace toxic chemical dispersants. Rocha e Silva et al. (2020) developed a sustainable plant-based biodetergent composed of a natural solvent (cotton seed oil), plant surfactant agent (saponin) and two natural stabilizers (carboxymethylcellulose and glycerin), which exhibited stability, efficiency (removing 100% of heavy oil from metallic surfaces) and the absence of toxicity. According to the authors, the application of the novel product would reduce both environmental impact and the risk posed to worker health related to toxic cleaning products. Arpornpong et al. (2020) investigated improved washing technology to treat drill cuttings from oil exploration and production sites with high concentrations of total petroleum hydrocarbons (TPHs). For such, a biologically based washing agent was formulated with a lipopeptide biosurfactant (in foam or cell-free broth), Dehydol LS7TH (fatty alcohol ethoxylate 7EO–an oleochemical surfactant) and butanol (as a lipophilic binder) in water. Due to the synergistic behavior between the anionic lipopeptides and the nonionic Dehydol LS7TH, the formula removed 92% of the TPHs from the drill cuttings when applied in a test jar. In a study conducted by Helmy, Gustiani & Mustikawat (2020), the application of biosurfactants in the formulation of a washing detergent was investigated and compared to that of a standard commercial detergent. The formulation comprising a mixture of a rhamnolipid, sodium tripolyphosphate as a builder and sodium sulphate as a filler was applied to wash stained cotton fabric. The results showed that the biosurfactant and its formulations are a promising substitute for synthetic counterparts, as for the results obtained in this work.

Therefore, biobased detergents are efficient and sustainable. In addition to promoting cleaning, these products are not toxic and have greater stability compared to conventional products. Biodetergent production will not generate any hazardous or environmentally harmful waste and does not rely on highly toxic raw materials. Thus, the use of green detergents is no longer merely a promise for industrial fields; it is a real possibility and an emerging necessity.

Conclusions

Advances in sustainable technologies have increasingly driven the search for natural biodegradable compounds that can achieve direct and indirect reductions in impacts on the environment, such as green formulations based on microbial surfactants. The biosurfactant from P. aeruginosa ATCC 10145 demonstrated viability for the formulation of a non-toxic detergent, with a high capacity to remove heavy oil impregnated on different surfaces. Thus, the present results demonstrate an innovative product that is superior in efficiency compared to chemical detergents currently on the market and is capable of reducing the environmental impacts arising from contamination by heavy oils. Since this is a laboratory study, viability for industrial application needs large-scale studies. Other parameters, such as aroma, color and long-term stability, also need to be considered. A future, more in-depth investigation with the on-site application of this novel product could indicate more applications in common industrial activities.

Supplemental Information

Supplemental Information 1 Surface tensions of P. cepacia CCT 6659 for 60 h at 250 rpm (28 and 37 °C) in canola frying oil and corn steep liquor, 96 h at 200 rpm at 28 and 37 °C in medium with glycerol and glucose (A) and in media with glucose (B).

Click here for additional data file.

Supplemental Information 2 Surface tensions of P. aeruginosa UCP 0992.

Fermentations performed for 96 h at 200 rpm at 28 and 37 °C in media containing sucrose (A) and glucose (B).

Click here for additional data file.

Supplemental Information 3 Surface tensions of P. aeruginosa ATCC 10145 and P. aeruginosa ATCC 9027.

Fermentations performed in media containing glycerol plus glucose, hexadecane plus glucose and molasses plus corn steep liquor at 28 °C (A) and 35 °C (B).

Click here for additional data file.

Supplemental Information 4 Emulsification capacity of biosurfactants from P. aeruginosa ATCC 10145 and P. aeruginosa ATCC 9027 cultivated in medium composed of 5.0% glycerol and 2.0% glucose for 96 h at 28 °C with different oils.

Click here for additional data file.

Supplemental Information 5 Emulsification capacity of biosurfactants from P. aeruginosa ATCC 10145 and P. aeruginosa ATCC 9027.

Click here for additional data file.

Supplemental Information 6 Dispersion of motor oil by biosurfactants from P. aeruginosa ATCC 10145 and P. aeruginosa ATCC 9027.

Click here for additional data file.

The authors are grateful to the technical support from the laboratories of the Icam Tech School of the Catholic University of Pernambuco (UNICAP) and the Advanced Institute of Technology and Innovation (IATI), Brazil.

Additional Information and Declarations

Competing Interests

Author Contributions

Data Availability

The authors declare that they have no competing interests.

Charles Bronzo B. Farias performed the experiments, analyzed the data, prepared figures and/or tables, authored or reviewed drafts of the paper, and approved the final draft.

Rita de Cássia F. Soares da Silva performed the experiments, analyzed the data, prepared figures and/or tables, authored or reviewed drafts of the paper, and approved the final draft.

Fabíola Carolina G. Almeida performed the experiments, analyzed the data, authored or reviewed drafts of the paper, and approved the final draft.

Valdemir A. Santos analyzed the data, authored or reviewed drafts of the paper, and approved the final draft.

Leonie A. Sarubbo conceived and designed the experiments, analyzed the data, prepared figures and/or tables, authored or reviewed drafts of the paper, and approved the final draft.

The following information was supplied regarding data availability:

The raw measurements are available in the Supplementary Files.

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
