# Peer review of "Removal of heavy oil from contaminated surfaces with a detergent formulation containing biosurfactants produced by Pseudomonas spp"

_PeerJ, doi:10.7717/peerj.12518_

## Round 0.1 · original submission · Major Revisions

Your manuscript needs major revisions.

·

Basic reporting

Basic data has been represented well by the authors.

Experimental design

Data on chemical characterization is missing
Interpretation of CMC in the form of plots are missing
Panel diagrams can b created to represent the emulsification of oils, carbon sources etc. It would help for the better comparison

Validity of the findings

It has good scientific data. However, just washing or dipping the oil coated material not sufficient to prove removal of oil. Some quantitative data is required which could be proved with other supporting experiments

Additional comments

Need to incorporate suggested data to improve the quality of the manuscript. It has been written well.

·

Basic reporting

The language is mostly fine and clear.
Background is not sufficient and hypothesis is not stated.

Experimental design

The study reports original data and the topic falls within the scope of the journal. However, the research question and knowledge gap are not clearly identified. The adopted methodology is standard for preliminary biosurfactants studies. Some experiments are not sufficiently presented with the necessary information. Replicates are included and statistical analysis was performed. Proper controls are lacking for some experiments.

Validity of the findings

Rationale of the study and benefit of the work to the literature are not clearly stated. Controls are lacking in some experiments.
Conclusions have some overstatements.

Additional comments

General Comments
This study deals with the production of biosurfactants by some Pseudomonas strains and investigates the potential of biosurfactants as components in a detergent formulation for removal of heavy oil from artificially contaminated metal, glass, and plastic surfaces. The manuscript is written in a clear and mostly correct language.
However, the study did not provide tangible insights into, or advancement of, our current understanding of the field. No dought, biosurfactants are highly interesting microbial products with tremendous applications in the environment and industry. Nonetheless, large-scale commercialization is still suffering from economic obstacles due to high production and purification costs. There is a major knowledge gap that needs further investigations towards enhancing the commercial viability of biosurfactants. The literature is full of reports on biosurfactants produced by Pseudomonas spp., and their physicochemical characteristics are very much similar to those reported in this study.
Perhaps the most interesting part of the study is the detergent formulation part. However, it is very truncated and many proper controls are lacking. I propose that the authors remove the initial experiments on the production of biosurfactants and dedicate a large-scale study with proper controls
to further explore the potential of biosurfactants-containing detergents.

Specific Comments
Title:
The title is too generic and can be modified to better reflect the performed work. For instance: “Removal of heavy oil from contaminated surfaces with a detergent formulation containing biosurfactants produced by Pseudomonas spp.”
Introduction:
This is one of the weak aspects of the study. The introduction presented general information on biosurfactants. The knowledge gap, rationale, and research questions, are not clear. The lack of several studies on biosurfactants-containing detergents does not necessarily make it a significant knowledge gap in the field. It is very important to elaborate on detergent formulations by answering the following questions.
-How detergent formulations are formed?
-What components do they have?
-What is the role of each component?
- Are they more efficient than using the surfactants alone? Why?
- Are there earlier studies on biosurfactants-containing detergents? Give examples.
- How they perform compared to conventional (synthetic detergents)?
- What are the knowledge gaps or questions in the topic?
- Which questions are addressed in this study and what is the approach?
It is fine to start the introduction with some general information in one or two paragraphs, albeit this should be followed by more specific presentation of the research problem and rationale of the study.
Materials and Methods:
Overall, the adopted methods are standard for biosurfactants studies and mostly described with sufficient detail. Replicates are included and statistical analysis is relevant. The main issue is the part describing the detergent formulations.
Line 116: What was the volumetric ratio of ethyl acetate to the culture broth?
How the product was concentrated?
Treatment with the base and acetone needs some details.
Line 128: Specify which liquid and which conc. of the biosurfactants.
Line 132: Which concentrations were used for CMC measurement?
Line 136: Why the emulsification activity was not measured for the extracted biosurfactants?
Line 150: The negative control should be uninoculated culture medium because the biosurfactants were in cell-free broth.
Line 159: The techno-economic feasibility was not investigated in this study.
Line 188: The designation “industrial detergent” is irrelevant.
Line 191-192: Was the biodegradability of the component assessed? The mentioned previous studies should be cited.
Line 193: The identity (chemical names and sources) of the natural organic solvent and the thickening agent should be revealed.
Line 198: Heating at which temp?
Line 202: This is not clear.
Line 203-206: This part needs some details on how the efficiency of the different formulations was evaluated? Which industrial field applications were investigated?
Line 244: Some of the physicochemical characteristic should be included.
Line 244-255: For feasible industrial application larger surfaces and higher amount of oil contamination should be tested. The glass slide and metal nuts with 100 μl oil do not reflect the application potential at industrial scale. What was the form of the biosurfactants preparation used in the detergent? Cell-free culture broth or crude extract. Was it added as solid or dissolved in a solvent?

Since the detergent formulation has several components, it is interesting to test them separately and in different combinations for oil removal. These controls should be included to reveal their effect (if any) on the oil removal efficiency of the biosurfactants.
Obviously, the biosurfactants should be tested separately for the oil removal. For all these controls, the different components should be in comparable concentrations. This is important to show whether the detergent has any benefit. If it does not have, then it makes no sense to use the biosurfactants in a detergent formulation to avoid further unnecessary costs.

Results:
Interpretations of the results should be moved to the discussion. For instance
Lines 317-321, Lines 393-404, Lines 334-343.
The quality of the figures should be improved in terms of resolution. For the bar charts, significance of data should be indicated by letters.
Line 400-401: Abrasion and lubrication were not investigated.
Line 402: Technical and environmental feasibility were not investigated.

Discussion:
The discussion is lengthy and, for the major part, cited literature without meaningful interpretation of the results.
The authors should elaborate on the observed variations in the properties of biosurfactants produced by different bacteria. Also, the differences in the performance of the detergent formulations need some explanation.
Line 132: The CMC is reported as mg/L

Conclusions:
The conclusion was presented more like a summary and focused only on the detergent formulation, although it represents a smaller part of the study. Moreover, some statements are not sufficiently supported with data. For instance, biodegradable (Line 565), industrial use (Line 566), organic pollutants (Line 569).
This is a lab study. Viability for industrial application needs large-scale or pilot studies.

---

## Round 0.2 · accepted · Accept

The reviewers were unable to re-review this revision but the manuscript was well improved and can be accepted for publication.